# Depth-Supervised Fusion Network for Seamless-Free Image Stitching

**Zhiying Jiang[1]**   **Ruhao Yan[2]**   **Zengxi Zhang[2]**   **Bowei Zhang[2]**   **Jinyuan Liu[2]***

[1] College of Information Science and Technology, Dalian Maritime University
[2] School of Software Technology, Dalian University of Technology
zyjiang0630@gmail.com    atlantis918@hotmail.com

## Abstract

Image stitching synthesizes images captured from multiple perspectives into a single image with a broader field of view. The significant variations in object depth often lead to large parallax, resulting in ghosting and misalignment in the stitched results. To address this, we propose a depth-consistency-constrained seamless-free image stitching method. First, to tackle the multi-view alignment difficulties caused by parallax, a multi-stage mechanism combined with global depth regularization constraints is developed to enhance the alignment accuracy of the same apparent target across different depth ranges. Second, during the multi-view image fusion process, an optimal stitching seam is determined through graph-based low-cost computation, and a soft-seam region is diffused to precisely locate transition areas, thereby effectively mitigating alignment errors induced by parallax and achieving natural and seamless stitching results. Furthermore, considering the computational overhead in the shift regression process, a reparameterization strategy is incorporated to optimize the structural design, significantly improving algorithm efficiency while maintaining optimal performance. Extensive experiments demonstrate the superior performance of the proposed method against the existing methods. Code is available at `https://github.com/DLUT-YRH/DSFN`.

## 1 Introduction

Image stitching is a fundamental task in computer vision, aiming to combine multiple images captured from different perspectives or positions into a single, high-resolution image with an extended field of view. This task plays a crucial role in various applications, such as panoramic photography [1, 2], remote sensing [3, 4], medical imaging [5, 6], and virtual reality [7, 8], where a comprehensive and seamless representation of a scene is required.

Conventional stitching methods predominantly follow a feature-driven paradigm, relying on hand-crafted local feature descriptors (e.g., SIFT [9], ORB [10]) for feature detection and robust matching algorithms (e.g., RANSAC [11]) to compute homography transformation matrices for reference and target images alignment. However, the planar scene assumption inherent to homography models fails to accommodate the complex geometric relationships arising from multi-depth layers in real-world scenarios, resulting in ghosting artifacts and structural misalignment in stitched results. To mitigate these issues, conventional methods adopt two primary optimization strategies: (1) enhancing local alignment accuracy through region-adaptive deformation techniques (e.g., mesh warping [12, 13, 14]), and (2) concealing residual artifacts by optimizing seam paths via energy functions [15, 16]. While effective in most scenarios, their performance is critically constrained by feature density and quality, leading to failures in low-texture, repetitive patterns, or large parallax scenarios.

---

*Corresponding author.

39th Conference on Neural Information Processing Systems (NeurIPS 2025).

Recent advancements in deep learning have introduced novel solutions for image stitching. Convolutional Neural Network (CNN)-based methods enable end-to-end learning of implicit geometric correlations between images, such as directly predicting transformation parameters through deep homography networks or modeling non-rigid deformations using deformable convolutions [17, 18]. Some works further integrate spatial attention mechanisms to enhance robustness against dynamic objects [19, 20] or employ unsupervised learning frameworks to address the scarcity of annotated real-world data [21]. Nevertheless, existing deep learning methods still face significant challenges: dependency on synthetic training data limits cross-domain generalization capabilities, while ensuring structural consistency in large-parallax scenarios remains challenging.

To address the aforementioned challenges, this paper proposes a depth-supervised image stitching that focuses on resolving co-planar alignment in large-parallax scenarios and ensuring seamless consistency transitions in multi-view overlapping regions. First, a two-stage depth-aware transformation estimation mechanism is introduced for large-parallax alignment. This mechanism leverages depth information to differentiate feature disparities of identical objects across varyi ng depth layers, while a recursive global-local deformation strategy integrates global homography estimation with localized adaptive warping, addressing the rigidity of conventional single-homography models in multi-plane scenes. During multi-view planar fusion, the optimal stitching seam is determined via graph-structured low-cost computation. A diffusion-based soft-seam propagation then generates pixel-wise confidence maps to define adaptive blending regions, effectively suppressing misalignment and ghosting artifacts caused by parallax. Additionally, we design a reparameterized strategy to optimize the shift regression model, ensuring the optimal effectiveness and the efficiency. The contributions are summarized as follows:

- We propose a depth-supervised image stitching, which focuses on addressing the alignment challenges caused by large parallax of significant depth differences, enabling the seamless fusion of multi-view images.

- The proposed method employs a depth-aware two-stage transformation estimation, coupled with a reparameterization strategy, which significantly enhances alignment performance in scenarios with large parallax.

- The determination of the soft-seam region enables a flexible adjustment for multi-view fusion, effectively avoiding issues such as misalignment and ghosting.

- Extensive experiments demonstrate that our method outperforms the state-of-the-arts, in terms of the large parallax alignment and seamless fusion.

## 2 Related Work

### 2.1 Image Stitching

**Feature-Based Image Stitching.** The core of manual feature-based image stitching lies in achieving accurate alignment through effective feature extraction and matching, which relies on sufficient geometric features in the scene. Brown *et al.* [22] pioneered this field by employing scale-invariant feature extraction combined with random sampling consistency to establish global rigid transformations. To address parallax issues, Li *et al.* [23] developed an analytical warp function based on point correspondences, enabling improved alignment through geometric constraints. Recognizing the limitations of single global transformations, Gao *et al.* [24] introduced dual-plane alignment by establishing separate warping models for distinct scene layers, though this approach faced challenges in complex environments with ambiguous planar divisions. Further advancing spatial adaptability, Zaragoza *et al.* proposed As Projective As Possible (APAP) [25], which localized mesh-based projective transformations, significantly increasing parameter flexibility while introducing artifacts at depth-discontinuous regions such as object boundaries.

The inherent alignment challenges in multi-view image stitching often manifest as ghosting artifacts within overlapping regions, necessitating sophisticated seam selection strategies. Zhang *et al.* [26] proposed a dual-scale alignment framework that preserves global structural consistency through optimal homography while enabling local seam-driven adjustments. Subsequent approaches focused on optimizing seam placement through energy minimization principles, with Kwatra *et al.* [27] introducing graph-based segmentation techniques to avoid object intersections. However, the computational intensity of these pixel-level optimization methods presents significant practical limitations

for real-world applications.

**Deep Learning-Based Image Stitching.** While contemporary feature descriptors [28, 29, 30] demonstrate potential for learned representations, their isolated application within traditional pipelines has limited practical adoption, driving research toward fully learned stitching frameworks. Deep learning approaches circumvent manual feature engineering by learning semantic representations through supervised (Lai *et al.* [19], Kweon *et al.* [20]), weakly supervised (Song *et al.* [31]), or unsupervised (Nie *et al.* [32]) paradigms, offering enhanced robustness in complex scenes. However, supervised methods' reliance on labeled data constrains their effectiveness in high-parallax scenarios. Nie *et al.* [21] pioneered unsupervised frameworks with improved cross-scene generalization and parallax tolerance, though persistent plane misalignments in extreme depth-varying scenes reveal fundamental limitations of current learning architectures.

## 2.2  Single Image Depth Estimation

Single image depth estimation aims to recover per-pixel depth from monocular visual data. Traditional approaches relied on geometric priors [33] or non-parametric depth transfer mechanisms [34], fundamentally constrained by color consistency assumptions. The advent of CNNs revolutionized this field through data-driven feature learning. Li *et al.* [35] pioneered multi-scale superpixel-to-pixel mapping via shallow CNNs with CRF refinement, though limited by local receptive fields. Liu *et al.* [36] advanced this by integrating CRF potentials within CNN frameworks, yet remained constrained by insufficient global context modeling. Eigen *et al.* [37] introduced a two-stage architecture that significantly enhanced spatial reasoning capabilities. Subsequent breakthroughs by Laina *et al.* [38] demonstrated the critical role of deep residual architectures in capturing holistic scene geometry through expanded receptive fields. The recent emergence of Depth Anything [39] marks a paradigm shift, establishing new state-of-the-art performance through unified representation learning.

## 3  The Proposed Method

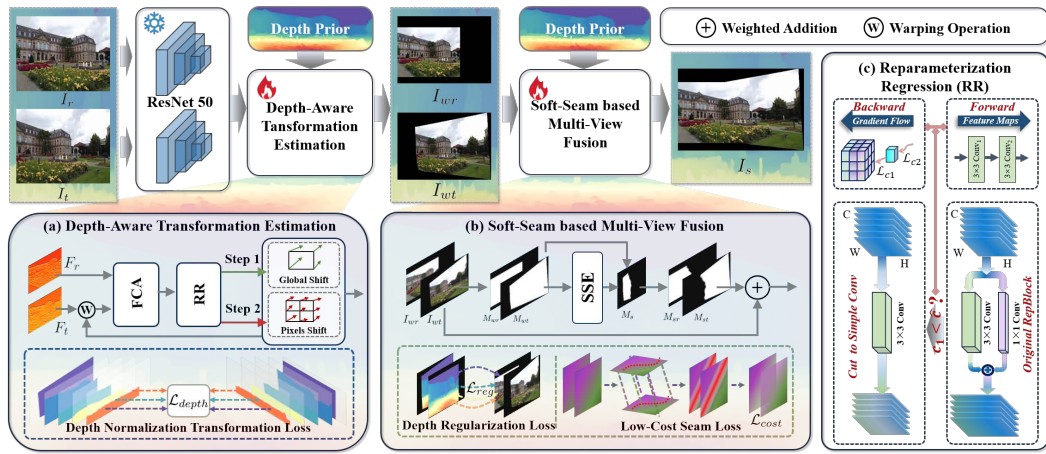

Figure 1: Workflow of the proposed method. It consists of two procedure: depth-aware transformation estimation and soft-seam based multi-view fusion. Besides, the transformation estimation process incorporates reparameterized regression to establish the optimal model.

As illustrated in Fig. 1, in the proposed method, we first feed the target image $I_t$ and reference image $I_r$ into the ResNet50 [38] for feature encoding. The extracted features from both views are then processed through a depth-aware transformation estimation module to obtain the warping matrices. To address alignment challenges in large parallax scenarios, a two-step recursive strategy is employed for the shift regression. Subsequently, the resulting transformations are applied to the observed images for alignment, and a soft-seam based multi-view fusion module is employed to blend the aligned images, producing a wide-field-of-view result $I_s$ with natural transitions and no visible artifacts. The comprehensive description of each module is provided in the following.

## 3.1 Depth-Aware Transformation Estimation

We employ ResNet50 [38] to initiate the multi-scale feature extraction from both reference and target images, generating feature pairs at $1/16$ and $1/8$ resolutions, denoted as $\{F_r^{1/16}, F_t^{1/16}\}$ and $\{F_r^{1/8}, F_t^{1/8}\}$. Beginning with the coarser $1/16$-scale features, the Feature Correlation Aggregation (FCA) block [40] computes inter-view correspondences through:

$$C_{i,j} = FCA(F_r^{1/16}, F_t^{1/16}), \tag{1}$$

where $C_{i,j}$ is the correlation volume. A regression block then predicts quadrilateral vertex offsets $\Delta p \in \mathbb{R}^{4 \times 2}$, from which a coarse homography matrix $H_C \in \mathbb{R}^{3 \times 3}$ is derived via Direct Linear Transformation (DLT) [41]:

$$H_C = \arg\min_H \sum_{k=1}^{4} \|p'_k - H \cdot p_k\|_2^2. \tag{2}$$

$p_k$ denotes the coordinate of the k-th point in the reference image, and $p'_k$ means the corresponding point of the target image. This initial alignment warps the target feature to $\hat{F}_t^{1/8} = H_C(F_t^{1/8})$. Subsequently, a mesh-based refinement stage employs grid-wise offset estimation for sub-pixel precision. Let $\mathcal{M} = \{(x_i, y_j)\}$ define the mesh grid, with Radial Basis Function (RBF) interpolation generating the continuous deformation field:

$$\Delta(x, y) = \sum_{m=1}^{M} w_m \phi(\|(x, y) - (x_m, y_m)\|), \tag{3}$$

where $\phi(r) = -e^{-(\epsilon r)^2}$ denotes the Gaussian basis function with shape parameter $\epsilon$. The final dense warping field $\mathcal{W}$ combines coarse homography and residual deformation:

$$\mathcal{W}(p) = H_C \cdot p + \Delta(p). \tag{4}$$

In the training process, we calculate the mean pixel error in the overlapping region after the coarse and residual transformations separately, which can be expressed as:

$$\begin{aligned}
\mathcal{L}_{alignment} =& f_{alignment}(I_r, I_t, \lambda, \gamma, \eta) \\
=& \lambda \|I_r \cdot M_H - \mathcal{W}_H(I_t)\|_1 + \\
& \gamma \|I_t \cdot M_{H^{-1}} - \mathcal{W}_{H^{-1}}(I_r)\|_1 + \\
& \eta \|I_r \cdot M_N - \mathcal{W}_\Delta(I_t)\|_1,
\end{aligned} \tag{5}$$

where $M_H$, $M_{H^{-1}}$ and $M_N$ are homography transformation masks, inverse homography transformation masks and residual masks, which are obtained through homography $H$, inverse homography $H^{-1}$, and residual transformation $\Delta$. $\lambda, \gamma, \eta$ are balance weights. For ease of calculation, we choose to transform the mask, and since nonlinear transformations do not always support inverse operations, we do not design inverse losses.

To preserve structural consistency with the original scene, we impose a shape-preserving constraint for the mesh. We design the mesh loss from the point of the edge size of a single mesh and the offset of adjacent meshes. The number of control points is recorded as $U \times V$, $\vec{e}_w$ and $\vec{e}_h$ are the set of two adjacent edges in the mesh, and the edge loss of a single mesh can be described as:

$$\mathcal{L}_{edge} = \frac{1}{U \times (V-1)} \sum_{\vec{e}_w} \sigma(\langle \vec{e}, \vec{i} \rangle - 2W_{mesh}) + \frac{1}{(U-1) \times V} \sum_{\vec{e}_h} \sigma(\langle \vec{e}, \vec{j} \rangle - 2H_{mesh}), \tag{6}$$

where $\vec{i}$ and $\vec{j}$ are unit horizontal and vertical vectors, $\sigma(\cdot)$ is a non-linear activation function, $H_{mesh}$ and $W_{mesh}$ are the length and width of a single mesh. By calculating the loss in the mesh, the mesh stretching is limited and the distortion is reduced. We believe that the adjacent edges between the meshes in the non-overlapping region should be as parallel as possible, so we constrain the mesh angle as:

$$\mathcal{L}_{angle} = \frac{1}{a} \sum_{\vec{e}_{e1}, \vec{e}_{e2}} \delta(1 - cos\theta), \tag{7}$$

where $a$ is the number of edge pairs, $\delta$ is the region label, and is denoted as 1 when the edge pair is in the non-overlapping region, 0 when the edge pair is in the overlapping region, and $\theta$ is the angle between the edge pairs. Considering the large parallax caused by significant depth variation, we incorporate the depth information as knowledge prior to supervise the learning of the transformation estimation. Specifically, we obtain the depth map through Depth Anything [39], characterizing the relative depth rather than the absolute depth. To this end, we perform normalization in the overlapping region of the reference and target images to reduce the relative error caused by the depth mutation of the non-overlapping region, expressed as:

$$\mathcal{L}_{depth} = f_{alignment}(I_{dr}, I_{dt}, \lambda', \gamma', \eta'). \tag{8}$$

where $I_{dr}, I_{dt}$ mean the depth maps of $I_r, I_t$. The total loss for depth-aware transformation estimation can be expressed as:

$$\mathcal{L}^t = \mathcal{L}_{alignment} + \mu\mathcal{L}_{edge} + \zeta\mathcal{L}_{angle} + \xi\mathcal{L}_{depth}. \tag{9}$$

### 3.2 Soft-Seam based Multi-View Fusion

Based on the results inferenced from the transformation estimation, we obtain the aligned image pair $I_{wr}, I_{wt}$. However, precise alignment remains challenging in real-world parallax scenarios, and the multi-view image fusion process must additionally ensure authenticity and accurate reconstruction, such as preserving structures and achieving natural transitions between multi-view scenes. To address this, we relax the conventional definition of "seams" in the stitching, and suppose that any region requiring fusion within overlapping areas can be treated as a potential seam. We build upon the low-cost seam localization and establish a soft-seam region diffused from the distinct seam to serve as the adaptive fusion adjustment, aiming to resolve the ghosting and misalignment artifacts while enabling natural transitions.

Specifically, we calculate the corresponding region masks $M_{wr}, M_{wt}$ based on the aligned images. These masks are fed into a Soft-Seam Estimation (SSE) to obtain soft-seam mask $M_s$ within overlapping areas, serving as the candidate region for fusion. SSE module is built upon a UNet architecture [42, 43], in which $3 \times 3$ convolutions are replaced with dilated convolutions, with dilation rates are set as $1, 2, 3, 4$, and $5$. At the four skip connections, the same-scale features from both input images are first upsampled using nearest-neighbor interpolation and then passed through a $1 \times 1$ convolution to reduce the number of channels. The difference map is first computed by subtracting the feature maps of the two images pixel by pixel. It is then concatenated with the upsampled features along the channel dimension, and the resulting representation is fed through two dilated convolution layers to further advance the decoding process.

$M_s$ is then integrated with the original aligned image masks through a single filter and the sigmoid function, yielding two more flexible masks $M_{sr}, M_{st}$ with pixel-level regional adaptability. These adaptive masks are subsequently applied to weighted fusion processing of the aligned images, enabling refined fusion tailored to local pixel characteristics.

In the training process, we first need to determine the terminal points of the seam, expressed as:

$$\mathcal{L}_{terminal} = \|(I_s - I_{wr}) \cdot (M_{wr} \odot \neg M_{wr})\|_1 + \|(I_s - I_{wt}) \cdot (M_{wt} \odot \neg M_{wt})\|_1. \tag{10}$$

It combines the inverted and original masks via element-wise multiplication to restrict the fusion mask boundary to the intersection area, controlling its endpoints. In which $\odot$ represents the pixel-by-pixel calculation of the two masks. $\neg M_{wr}$ and $\neg M_{wt}$ represent the inverted and expanded mask. In order to improve the sensitivity of the difference values, we choose the pixel square difference to construct the cost map and the cost loss is defined as:

$$\mathcal{L}_{cost} = \sum_{i,j} |M_s^{i,j} - M_s^{i+1,j}|(D^{i,j} + D^{i+1,j}) + \sum_{i,j} |M_s^{i,j} - M_s^{i,j+1}|(D^{i,j} + D^{i,j+1}), \tag{11}$$

where $D$ is the squared difference between the warped image $I_{wr}$ and $I_{wt}$. In order to constrain the smoothness of the fused image, the smoothness loss, which calculates the smoothness penalty by measuring the distance between adjacent pixels within the fusion region of the stitched image, is also adopted:

$$\mathcal{L}_{smooth} = \sum_{i,j} |M_s^{i,j} - M_s^{i+1,j}|(I_s^{i,j} - I_s^{i+1,j}) + \sum_{i,j} |M_s^{i,j} - M_s^{i,j+1}|(I_s^{i,j} - I_s^{i,j+1}). \tag{12}$$

Furthermore, we also introduce the depth consistency to supervise the inference results. Specifically, the base depth maps are first aligned with the estimated transformation. Then, a secondary local regularization of the aligned depth images $I_{wdr}, I_{wdt}$ is performed to further calibrate the local relative depth, expressed as:

$$\mathcal{L}_{reg} = \sum_{i,j} |M_s^{i,j} - M_s^{i+1,j}|(\mathcal{F}(I_{wdr}, I_{wdt})^{i,j} - I_{wdt}^{i+1,j})$$
$$+ \sum_{i,j} |M_s^{i,j} - M_s^{i,j+1}|(\mathcal{F}(I_{wdr}, I_{wdt})^{i,j} - I_{wdt}^{i,j+1}). \quad (13)$$

$\mathcal{F}$ denotes the fusion process. The total loss for the soft-seam based multi-view fusion can be expressed as:

$$\mathcal{L}^f = \rho\mathcal{L}_{terminal} + \tau\mathcal{L}_{cost} + \iota\mathcal{L}_{smooth} + \sigma\mathcal{L}_{reg}. \quad (14)$$

### 3.3 Reparameterization Regression

In the realm of learning-based image stitching methodologies, the adoption of fully connected architectures for shift regression often incurs significant computational costs while yielding suboptimal performance. To mitigate this issue, we leverage reparameterization techniques [44] to identify the optimal structural configuration during the parameter regression process.

Although reparameterization techniques enhance feature diversity and structural flexibility through the introduction of Reparameterization Blocks (RepBlocks), existing reparameterized architectures fail to achieve robust performance improvements due to inherent limitations in RepBlocks [45]. To address this, we propose a Reparameterization Block Adaption (RBA) algorithm, which dynamically adapts the model training process by selectively integrating either a RepBlock or a standard convolutional layer based on the specific requirements of the convolution layer.

Specifically, in the model forward process, different convolution structures provide distinct feature representations to extract diverse feature maps from input features. Therefore, in the initial model, following the research in [46], we replace the $3 \times 3$ convolution in our regression block located behind the FCA block of the proposed depth-aware transformation estimation to a RepBlock consists of a $1 \times 1$ convolution layer $Conv^1$ and a $3 \times 3$ convolution layer $Conv^3$. To evaluate the contribution of these two layers, we formulate a linear combination. Given a set of input feature maps $f_{in} \in \mathbb{R}^{C_{in} \times H \times W}$, where $C_{in}$, $H$ and $W$ are the input channel number, height and width, the output features $f_{out}$ of a RepBlock are calculated as:

$$f_{out} = Re(\mathbf{w_1} \times Conv^1(f_{in}) + \mathbf{w_3} \times Conv^3(f_{in}) + \mathbf{b}), \quad (15)$$

where $\mathbf{w_1}$, $\mathbf{w_3}$ and $\mathbf{b} \in \mathbb{R}^{C_{out} \times 1 \times 1}$ are the weights and bias. $Re$ denotes the ReLU function. $f_{out} \in \mathbb{R}^{C_{out} \times H \times W}$, where $C_{out}$ is the output channel number. In our formulation, $\mathbf{w_1}$ and $\mathbf{w_3}$ can be trained to evaluate the contribution of the two branches. The contribution $c_1$ of the $Conv^1$ in the RepBlock is calculated as:

$$c_1 = \frac{\frac{1}{C_{out}}(\sum \mathbf{w_1})}{\frac{1}{C_{out}}(\sum \mathbf{w_1}) + \frac{1}{C_{out}}(\sum \mathbf{w_3})}. \quad (16)$$

The effect of our RBA is to prevent the output features of the $Conv^1$ layer from playing a damaging role in feature extraction. Therefore, in the training process, if $c_1 < \hat{c}$, where $\hat{c}$ presents the hyperparameter of the minimum threshold, we cut the $Conv^1$ by coupling it to the $Conv^3$ layer, and the parameter weight $\mathbf{W}_3^{new}$ of the new $3 \times 3$ layer $Conv^{3,new}$ is calculated as:

$$\mathbf{W}_3^{new} = \mathbf{w_3} \cdot \mathbf{W_3} + \mathbf{w_1} \cdot pad(\mathbf{W_1}), \quad (17)$$

where $\mathbf{W_1}$ and $\mathbf{W_3}$ are the weights of $Conv^1$ and $Conv^3$, and the $pad$ operation indicates adding value 0 around the $\mathbf{W_1}$ [44]. After cutting the $1 \times 1$ branch, we ensure the model can be adapted to the training task while avoiding the feature degradation caused by the additional branches.

## 4 Experiments

### 4.1 Implementation Details

Our method is implemented using the PyTorch framework and executed on an NVIDIA RTX 3090 GPU. For training both the depth-aware transformation estimation and soft-seam based multi-view

fusion models, we employ the Adam optimizer [47], and the learning rate decays exponentially, with an initial value of $10^{-4}$. The transformation model is trained for 100 epochs, with the hyperparameters $\lambda$, $\gamma$, and $\eta$ set to 3, 3, and 1, respectively. The values of $\lambda'$, $\gamma'$, $\eta'$ are identical to those of $\lambda$, $\gamma$, and $\eta$. $\mu$, $\zeta$, $\xi$ are set to 10, 10 and 0.3. For the multi-view fusion model, we initially train the model for 50 epochs on the training set, with the hyperparameters $\rho$, $\tau$, $\iota$, $\sigma$ set to 10000, 1000, 1000, 10.

The UDIS-D training set [32] is employed as the training data. To enhance the reliability of the experimental results, we evaluate the model on the UDIS-D testing set and further validate it using real-world data from the IVSD dataset [40].

## 4.2 Performance Comparison

We compare our method with APAP [25], ELA [23], LPC [48], SPW [49], UDIS [32], UDIS++ [21], TRIS [50] and SRS [51], where each method adopts the pre-trained model and configuration parameters provided by the official.

### 4.2.1 Qualitative Evaluation

The qualitative comparison on the UDIS-D dataset is shown in Fig. 2. In the first example, our method achieves a clearer fusion result for the wall area, effectively avoiding issues such as blurring and ghosting artifacts. Additionally, within the region marked by the red framework, the proposed method successfully preserves the complete information of the bicycle without any loss of content. In the second example, while other comparative methods exhibit ghosting or content loss, our method delivers a stitching result that is both visually clear and realistic, demonstrating a better reconstruction quality. Furthermore, to compare the alignment accuracy of different methods, we visualize the alignment errors in the lower-right corner of the corresponding figures. It can be observed that the proposed method achieves significantly higher alignment precision compared to the others.

Visual results on the IVSD dataset are presented in Fig. 3. The proposed method demonstrates superior stitching performance across varying depths of the captured scene, with alignment errors further confirming its effectiveness. The consistent performance across both datasets robustly validates the efficacy of the proposed method.

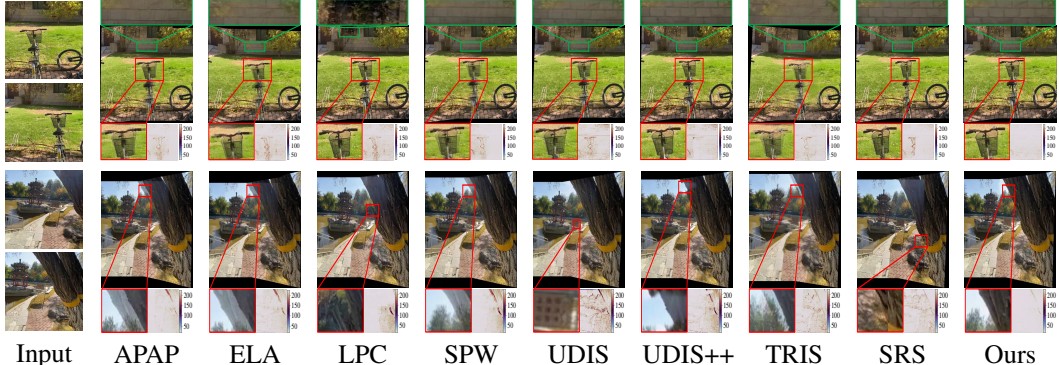

Input   APAP   ELA   LPC   SPW   UDIS   UDIS++   TRIS   SRS   Ours

Figure 2: Visual comparison of stitched images from UDIS-D dataset. The alignment error is visualized in the lower right corner.

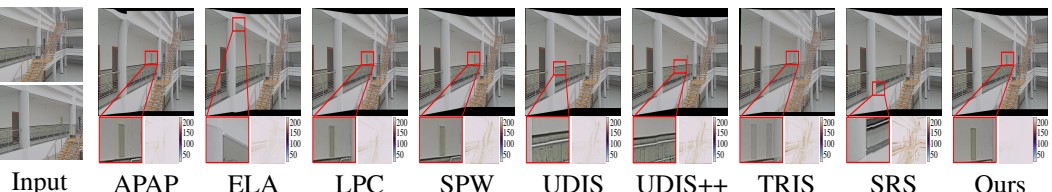

Input   APAP   ELA   LPC   SPW   UDIS   UDIS++   TRIS   SRS   Ours

Figure 3: Visual comparison of stitched images from IVSD dataset. The alignment error is visualized in the lower right corner.

Table 1: Quantitative comparison on UDIS-D and IVSD datasets. The best and second results are marked in **red** and **blue**.

| Method | UDIS-D | | | | IVSD | | | |
|---|---|---|---|---|---|---|---|---|
| | PSNR(↑) | SSIM(↑) | SIQE(↑) | LPIPS(↓) | PSNR(↑) | SSIM(↑) | SIQE(↑) | LPIPS(↓) |
| APAP [25] | 23.792 | 0.794 | 41.707 | 0.472 | 22.904 | 0.681 | 39.281 | 0.454 |
| ELA [23] | 24.012 | 0.808 | 41.781 | 0.470 | 23.452 | 0.701 | 37.186 | **0.435** |
| LPC [48] | 22.595 | 0.736 | **43.616** | 0.467 | 20.996 | 0.641 | 37.517 | 0.447 |
| SPW [49] | 21.606 | 0.687 | 41.060 | **0.466** | 18.868 | 0.575 | 36.156 | 0.449 |
| UDIS [32] | 21.171 | 0.648 | 42.186 | 0.475 | 23.535 | 0.743 | 40.474 | 0.451 |
| UDIS++ [21] | **25.426** | **0.837** | 43.184 | 0.469 | **26.649** | **0.819** | **46.383** | 0.439 |
| TRIS [50] | 24.476 | 0.821 | 41.621 | 0.476 | 24.187 | 0.753 | 40.873 | 0.448 |
| SRS [51] | 24.828 | 0.811 | 41.857 | 0.473 | 24.234 | 0.796 | 35.641 | 0.445 |
| Ours | **25.467** | **0.839** | **43.732** | **0.462** | **26.778** | **0.820** | **46.568** | **0.436** |

Table 2: Efficiency comparison against the state-of-the-art methods.

| Methods | APAP [25] | ELA [23] | LPC [48] | SPW [49] | UDIS [32] | UDIS++ [21] | TRIS [50] | SRS [51] | Ours |
|---|---|---|---|---|---|---|---|---|---|
| Time (ms) | 6683.14 | 8347.79 | 13435.47 | 11651.68 | 193.66 | 79.73 | 107.98 | 83.17 | 67.04 |

#### 4.2.2 Quantitative Evaluation

We employ a set of evaluation metrics, including PSNR [52], SSIM [53], SIQE [54], and LPIPS [55], to conduct a comprehensive performance assessment. According the UDIS++ [21], the test sets of UDIS-D are categorized into three levels based on their complexity, and the corresponding quantitative results are summarized in Table 1. Furthermore, to rigorously evaluate the generalization capability of the proposed method, quantitative results on the IVSD dataset are also presented in Table 1. It is evident that the proposed method outperforms other methods across multiple metrics, substantiating its superiority further.

To further evaluate the efficiency of the proposed method, we present the processing time required by each comparative method for image stitching at a size of $512 \times 512$, with the results summarized in Table 2. It can be observed that, although our method involves deep estimation and inference processes, the overall running time is still better than that of the other methods, making it more suitable for practical applications.

#### 4.2.3 User Study

To evaluate the subjective performance of our method, we conducted a user study to assess the visual quality of the stitched images. The input and output images were organized according to their scene categories, and participants were presented with a set of input images alongside the corresponding output images generated by each comparison method. Participants were asked to evaluate quality from multiple perspectives, including ghosting, misalignment, structural accuracy, and realistic scene restoration. They were allowed to zoom in or out for detailed observation and were instructed to rate each image on a scale of 1 to 5 based on visual quality. The study involved 50 participants, including 30 researchers or students with a background in computer vision and 20 individuals without specific expertise. The results of the user study, as illustrated in Fig. 4, demonstrate that our method consistently received higher ratings compared to other methods.

### 4.3 Ablation Studies

**Loss Function Evaluation.** We conduct a comprehensive ablation study to analyze the impact of different constraints in the depth-aware transformation estimation and soft-seam based multi-

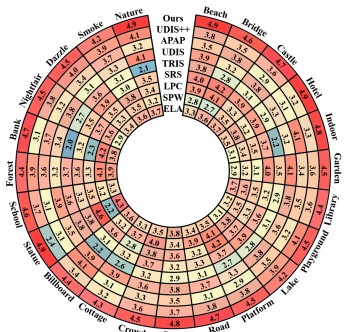

Figure 4: Visual quality survey.

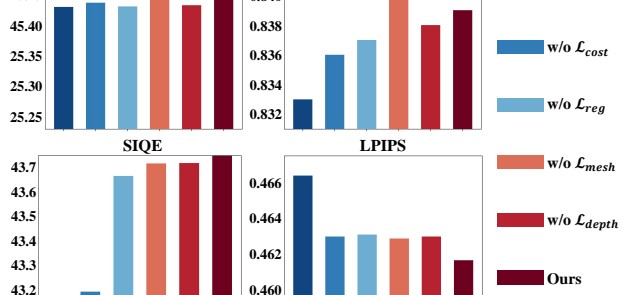

Figure 5: Ablation study on different loss components.

view fusion modules, respectively. During the experiments, we would like to clarify that $\mathcal{L}_{edge}$ and $\mathcal{L}_{angle}$ exhibit complementary effects, where retaining one constraint alone renders the other nearly ineffective. Therefore, we combine these constraints into a unified loss term, denoted as mesh loss $\mathcal{L}_{mesh}$ for ablation analysis. As shown in Fig. 6, the removal of mesh constraints in the transformation estimation leads to noticeable distortions in the deformed images, while the absence of fusion module constraints hinders the model's ability to compensate for stitching errors caused by insufficient local alignment. Furthermore, the lack of depth supervision not only increases the visual errors in both transformation and fusion but also degrades the overall accuracy of the method. The quantitative results across the entire UDIS-D dataset are presented in Fig. 5 and Table. 3. Although the ablation of mesh constraints marginally improves certain metrics by relaxing the image distortion limits, this improvement is not meaningful from a visual perspective.

**Soft-Seam Fusion Evaluation.** We also perform an ablation study of the fusion strategy compared with average fusion and seam cutting [27]. As shown in Fig. 7, the first row exhibits the gradient results from different fusion strategies and the second row visualizes the corresponding fusion regions. We can see that a larger fusion region increases the likelihood of ghosting artifacts. However, an excessively small fusion region may result in insufficient gradient smoothness. The proposed soft-seam fusion strategy adaptively preserves gradient continuity to the greatest extent, facilitating natural and seamless results.

**Adaptive Mask Evaluation.** The fusion mask in our method is derived from an adaptive weight matrix based on the soft-seam region. A comparison with the absolute weight derived from traditional seam-based methods is illustrated in Fig. 8. The result produced by our method exhibits a more visually appealing effect and smoother region.

**Reparameterization Evaluation.** We test 10 thresholds for hyperparameter selection, where $\hat{c} = 1$ represents the original model and $\hat{c} = 0$ represents the model without RBA. Results shown in Fig. 9 indicate that both the original model and the full reparameterization model cannot perform the best. The model with $\hat{c} = 0.25$ realizes the best performance, while maintaining efficient training.

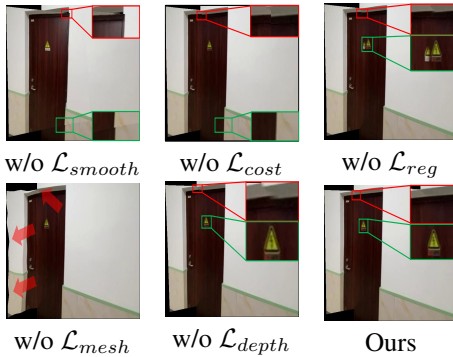

Figure 6: Ablation study on loss function.

| Loss | PSNR(↑) | SSIM(↑) | SIQE(↑) | LPIPS(↓) |
|---|---|---|---|---|
| w/o $\mathcal{L}_{smooth}$ | 25.431 | 0.833 | 43.156 | 0.466 |
| w/o $\mathcal{L}_{cost}$ | 25.438 | 0.836 | 43.186 | 0.463 |
| w/o $\mathcal{L}_{reg}$ | 25.432 | 0.837 | 43.651 | 0.463 |
| w/o $\mathcal{L}_{mesh}$ | **25.473** | **0.840** | 43.701 | **0.463** |
| w/o $\mathcal{L}_{depth}$ | 25.434 | 0.838 | **43.703** | 0.463 |
| Ours | **25.470** | **0.839** | **43.732** | **0.462** |

Table 3: Ablation study on loss components of transformation estimation and multi-view fusion models.

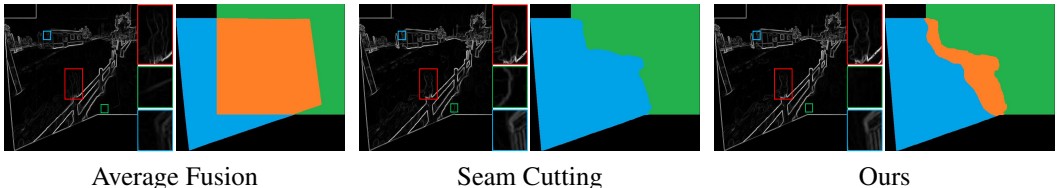

| Average Fusion | Seam Cutting | Ours |

Figure 7: Ablation study of the fusion strategy. The corresponding fusion regions are visualized on the right part.

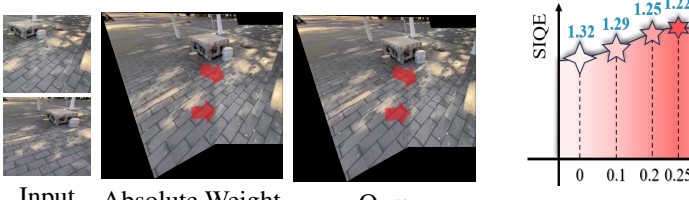

Input    Absolute Weight    Ours

Figure 8: Visual comparison between the proposed adaptive mask and the absolute mask.

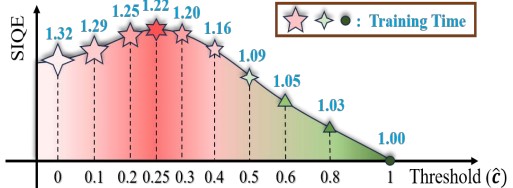

Figure 9: Ablation study on the hyperparameter $\hat{c}$ in the RBA.

## 5 Limitations

The proposed method is primarily designed for stitching two images and currently lacks full capability to address the challenges associated with multi-image panoramic stitching. Key limitations include difficulties in maintaining loop consistency and mitigating global error propagation in complex scenarios. These issues can compromise the geometric coherence of the final output, thereby constraining the method's robustness and broader applicability.

## 6 Conclusion

This paper proposed a depth-supervised image stitching method designed to address the alignment challenges in large parallax scenarios and achieve seamless wide field-of-view reconstruction. Firstly, a depth-aware two-stage transformation estimation is developed, which leverages depth-consistency priors to align targets across varying depth ranges. Secondly, a soft-seam region diffusion strategy is introduced to accurately identify transition regions, enabling natural and smooth fusion while mitigating ghosting and misalignment issues. Additionally, the reparameterization strategy for shift regression enhances the adaptability and reduces computational overhead. Extensive experiments validate the effectiveness of the proposed method. Although our method can improve multi-view alignment and fusion performance by leveraging depth consistency guidance, the presence of dynamic elements in the scene poses challenges for obtaining accurate depth information. In the future, we will focus on robust stitching under dynamic conditions to further enhance alignment robustness in such scenarios.

## 7 Acknowledgment

This work was supported in part by the National Natural Science Foundation of China under Grant 62302078, in part by the Fundamental Research Funds for the Central Universities under Grant 3132025276, and in part by China Postdoctoral Science Foundation under Grant 2023M730741.

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
