# OpenReview forum: "Depth-Supervised Fusion Network for Seamless-Free Image Stitching"
_NeurIPS.cc/2025/Conference — NeurIPS 2025 poster_

### Official Review · Reviewer_zXpV · 2025-06-30

**Clarity:** 3
**Significance:** 3
**Originality:** 3
**Rating:** 6
**Confidence:** 5

**Summary:**

This paper presents a novel two-stage image stitching method enhanced with deep supervision, targeting the challenge of stitching images with significant parallax. By combining linear and nonlinear deformations alongside soft seam region fusion, the method achieves high-quality, seamless image stitching. Reparameterization further boosts computational efficiency. Experimental results validate the effectiveness of this method.

**Questions:**

1. Does the method proposed in this paper have a strong dependence on the generation quality of depth maps? Will the absence of an effective depth map lead to errors in the stitched image?
2. Why is the method of combining linear transformation and nonlinear transformation adopted for image alignment? Have you ever attempted to directly achieve alignment through nonlinear deformation?
3. In the ablation experiment, why do some indicators show some improvement when the mesh constraint is removed? Does it mean that the mesh constraint is meaningless?

**Ethical Concerns:**

["NO or VERY MINOR ethics concerns only"]

**Final Justification:**

Through the authors' rebuttal, my concerns have been well addressed. The proposed depth-supervised image fusion method demonstrates excellent fusion performance and good generalization ability, and it is novel to the image fusion field.

**Limitations:**

Yes.

**Paper Formatting Concerns:**

N/A.

**Quality:**

4

**Strengths And Weaknesses:**

Strengths:
1. This paper presents a two-stage image stitching method designed to achieve high-quality stitching of large parallax images. This method introduces deep supervision to address the challenges associated with aligning images that have significant differences in perspective and depth.
2.  Reparameterization is employed to enhance the computational performance of the stitching process.  This technique not only improves the efficiency of the method but also reduces the consumption of computing resources, making it more practical for real-world applications.
3. This paper also incorporates depth information into the image stitching.  By leveraging depth cues, the method optimizes the stitching of large parallax images, leading to more accurate and visually pleasing results.

Weakness:
The manuscript presents a multifaceted evaluation of the proposed method. However, the visual comparisons are limited, and the robustness under different imaging conditions requires further investigation. Additional experimental results are recommended.

---

> ### Author Rebuttal · Authors · 2025-07-30
>
> ## Summary of the reviews
>
> We appreciate the recognition of our paper’s strengths in: (1) a two-stage image stitching method to handle large parallax and depth differences, (2)  reparameterization strategy, and (3) depth constraint on challenging images. We acknowledge the reviewer's concerns about (1)generalization, (2) depth dependence, (3) linear-nonlinear transformation estimation (4) constraint improvement. We will address these concerns in the following.
>
> ### Weakness: Generalization
> **Response:** Thank you for your suggestion. Due to page limitations, we have provided more visual comparison results in the supplementary materials. The input images cover a wide range of environments, including nighttime, low-light conditions, large wall surfaces, expansive ground areas, and scenes with large depth of field. In the future, we will also upload the supplementary materials alongside the main text to allow for a more comprehensive performance comparison.
>
> ### Question 1: Depth Dependence
> **Response:** We would like to clarify that our method utilizes depth information as a constraint to improve multi-view alignment accuracy, and exploits soft seam localization to enable reliable wide-field image reconstruction. While high-quality depth maps can facilitate the production of better stitching results, they are not the sole determining factor for the effectiveness of our method.
>
> During the transformation estimation stage, we leverage the relative depth variations between objects in the image to guide more precise alignment, as depth differences may exist among objects. In the fusion stage, since the depth variation within a single object tends to be regular, we extend the stitching seam to create a flexible space and perform adaptive fusion to achieve accurate wide-field reconstruction.
> Therefore, our method is fundamentally guided by depth maps to enable improved image stitching, rather than relying on depth maps to directly complete stitching. The combination of depth constraint and soft-seam localization ensures robust and high-quality stitching performance.
>
>
> ### Question 2:  Linear-nonlinear Transformation Estimation
> **Response:** Linear transformations facilitate global and consistent deformation of the image, helping to preserve the overall structure and maintain shape integrity. Nonlinear transformations, meanwhile, allow for more flexible local adjustments, which can improve the alignment of fine details in multi-view scenarios. While directly applying more flexible nonlinear transformations for multi-view alignment may seem intuitive, we find that relying solely on nonlinear transformations achieves good alignment only when the parallax between images is small. When there is significant parallax, this approach can easily lead to a loss of shape preservation, resulting in stretching or distortion and thus a decline in visual quality. Therefore, we adopt a coarse-to-fine strategy that combines linear and nonlinear transformations for image alignment. Specifically, we first use linear transformation for global initial alignment, and then apply nonlinear transformation to locally refine the result. This method ensures the overall structural consistency while also improving the alignment accuracy in local regions.
>
> ### Question 3: Constraint Improvement
> **Response:** In our method, the mesh constraint is introduced to prevent excessive stretching and to preserve the intrinsic shape of the mesh. As illustrated in Fig. 6 of the submitted manuscript, removing the mesh constraint (denoted as w/o mesh) can improve alignment in the overlapping regions, but it also lessens the effectiveness of control point displacement, resulting in wave-like and irregular distortions at the image boundaries. Due to the unavailability of ground truth for stitched images in real-world scenarios, previous works typically evaluate PSNR and SSIM metrics only within overlapping regions, rather than across the entire image. This explains why the w/o mesh configuration achieves the best performance on these metrics. In contrast, the SIQE and LPIPS metrics consider image quality on a global scale, and our method exhibits clear advantages on these global metrics. Taken together, both the quantitative and qualitative results demonstrate that the design of our method is reasonable and effective.

---

> > ### Comment · Reviewer_zXpV · 2025-08-04
> >
> > Thank you for your efforts, and my concerns have been addressed. However, I hope the authors can further provide the running time of the proposed method, which will make this work more comprehensive.

---

> > > ### Author Response · Authors · 2025-08-05
> > >
> > > We thank the reviewer for their positive feedback and valuable suggestions.
> > >
> > > Regarding the running time, our method takes approximately 67.04 ms to process stitching for images with a size of 512×512. To further validate the efficiency of our algorithm, we compared the running times of different methods in the table below. It can be observed that, although our method involves deep estimation and inference processes, the overall running time is still better than that of the other methods, making it more suitable for practical applications.
> > > | APAP  | ELA   | LPC   | SPW   | UDIS  | UDIS++ | TRIS  | SRS   | **Ours** |
> > > |-------|-------|-------|-------|-------|--------|-------|-------|----------|
> > > | 6683.14ms| 8347.79ms| 13435.37ms | 11651.68ms| 193.66ms| 79.73ms | 107.98ms| 83.17ms| 67.04ms |
> > >
> > > We will include this information, along with the comparative results, in the revised manuscript to make the work more comprehensive. Hope this addresses your concerns.

---

> > > > ### Comment · Reviewer_zXpV · 2025-08-05
> > > >
> > > > Thank you for your prompt reply. I have no further concerns now, and I recognize the value of this work to the image fusion community.

---

### Official Review · Reviewer_Hpo2 · 2025-06-30

**Clarity:** 1
**Significance:** 2
**Originality:** 3
**Rating:** 4
**Confidence:** 4

**Summary:**

This work aims for image stitching at challenging scenes with low textures and large parallax. The overall pipeline is consists of two stage; transformation estimation and multi-view fusion stage. At the transformation stage, image features from ResNet50 are aggregated with attention modules, and the regression network estimates the DLT parameters for coarse warping and a residual deformation field.
In this stage, the authors newly suggested the mesh loss (edge + angle) which preserves the shape consistency of the mesh, and the depth loss which reduces the difference between relative depths earned from DepthAnything in the overlapping region.
At the fusion stage, the authors proposed a soft-seam estimation module, which outputs soft-seam region from the mask of warped target and reference images. Depth values are again regularized to have similar values in warped pixels at this stage.
Lastly, authors suggested to use RepBlock (RepVGG) at the regression block of the transformation estimation stage. Instead of naively incorporating the RepVGG layer, they proposed a reparameterization algorithm that selectively integrates the RepBlock or a standard convolution layer.
The quality of the proposed method was evaluated on the UDIS-D dataset through quantitative, qualitative, and user studies.                    . 	`

**Questions:**

1. The alignment of two depth maps without camera positions of two views can not be trivially done with 2D warping. The minimum and maximum values taken for depth normalization should be carefully chosen depending on the distance between the surface and the camera, to make warped affine depth maps to be numerically similar. However, the paper does not include the depth normalization method in both the transformation estimation module and the fusion module. In ablation, there was no significant quality enhancement by the depth loss and the regression loss. I doubt the correctness and effectiveness of depth supervision modules.

2. Most of the network architecture specifications are excluded in this paper. 1) There is no exact explanation of the soft-seam estimation module’s operation. 2) It’s hard to find where the RBA algorithm was adopted since the regression module's architecture was not described.

3. The quantitative improvements by the regularizations are numerically weak (Table 2) compared to the noticeable quality enhancement in Figure 6. Please analyze this mismatch and supplement the effectiveness of the suggested contributions.

4. What is the inherent limitation of RepBlocks? (line 193). Please explain the specific behavior. Why does it need to be selectively incorporated instead of completely removing it?

**Ethical Concerns:**

["NO or VERY MINOR ethics concerns only"]

**Final Justification:**

After reading the other reviewers' comments and the authors' reply, my concerns regarding the clarification of contributions and their effectiveness have been resolved. Therefore, I am willing to raise my score to borderline accept.

**Limitations:**

Yes

**Quality:**

2

**Strengths And Weaknesses:**

Strength
1. The authors adopted a monocular depth estimation network to resolve the large parallax caused by the big depth difference. The authors suggested a regularization loss that can incorporate prior depth knowledge into the end-to-end image stitching pipeline.
2. The authors suggested a novel approach to consider seam as a candidate area and adopted a soft seam estimation module to estimate the blending regions. The qualitative result of ablation shows that such soft blending achieves a sharp yet smooth transition in the image gradient space.

Weakness.
1. In the transformation estimation stage, it’s hard to differentiate the contribution of this work and the contribution of previous studies (FCA block and alignment loss). Dividing the sections into preliminaries and the contribution would be helpful.
2. There is no explanation of the operations or the architecture of the Soft-Seam-Estimation module which is the key contribution. The paper only demonstrates the input and output. Moreover,  the loss equations (10), (11), and (12) are not intuitive enough to understand their objective in the optimization. What kind of pixel calculation does ⊙? Why does this operation work on the inverted and non-inverted region mask?
3. In the ablation study (Figure 5) the result of the full model does not excel in most evaluation metrics except SIQE. This violates the argument that adopting the regularization and depth prior will enhance the stitching quality. Is there a reason for the mismatch between qualitative results and quantitative results?

---

> ### Author Rebuttal · Authors · 2025-07-30
>
> ## Summary of the reviews
>
> We appreciate the recognition of our paper’s strengths in: (1) depth prior constraints, (2)  soft-seam candidate fusion, and (3) comprehensive ablation analysis. We acknowledge the reviewer's concerns about (1) the innovativeness of the transformation estimation, (2) the architectural design of the soft-seam fusion, (3) the necessity of the reparameterization, and (4) the inconsistency between quantitative and qualitative results. We will address these concerns in the following.
>
> ### Weakness 1: Contribution of the Proposed Transformation Estimation
> **Response:** We would like to clarify that previous works perform correlation computations on image features and train the transformation network using self-supervised constraints between pre-aligned and post-aligned images. In these works, the reference information provided by image features is quite limited, especially in scenes with repetitive content, such as large surfaces of ground or buildings. In such cases, similar texture regions often exhibit significant depth variations. Relying solely on the appearance features for transformation estimation makes it difficult to accurately establish correspondences between different viewpoints. To this end, we introduce a regularized depth constraint, enabling more precise multi-view alignment by jointly constraining content and depth information in multiple dimensions. Furthermore, to enhance feature diversity, we employ a reparameterization strategy to provide additional frequency features, which further strengthens the representation of image features and leads to more accurate transformation estimation results.
>
> Based on your feedback, we would place greater emphasis on highlighting the differences between the proposed method and previous works to further underscore the contributions of our method.
>
> ### Weakness 2: Architecture and Equations
> **Response:** The Soft-Seam Estimation module is designed to predict a pixel-level adaptive weighted fusion mask within the overlapping region of two aligned images, enabling fine-grained fusion based on local pixel features. This module is built upon a UNet architecture, in which 3×3 convolutions are replaced with dilated convolutions, with dilation rates are set as 1, 2, 3, 4, and 5. At the four skip connections, the same-scale features from both input images are first upsampled using nearest-neighbor interpolation and then passed through a 1×1 convolution to reduce the number of channels. The difference map is then obtained by subtracting the features from the two images pixel by pixel. This difference map is concatenated with the upsampled features along the channel dimension and fed into two layers of dilated convolutions (with the same dilation rate), continuing the decoding process. In this way, the inconsistency information of the overlapping region is explicitly injected into the network, enabling more precise seam localization.
>
> Eq (10) combines the inverted and original masks via element-wise multiplication to restrict the fusion mask boundary to the intersection area, controlling its endpoints. ⊙ denotes element-wise multiplication.
> Eq (11) computes the squared Euclidean distance between corresponding pixels of the two warped images within the fusion region.
> Eq (12) calculates the smoothness penalty by measuring the distance between adjacent pixels within the fusion region of the stitched image.
>
> ### Weakness 3: Quantitative Result Analysis
> **Response:**  In the ablation study, we evaluated performance using PSNR, SSIM, SIQE, and LPIPS. Higher PSNR, SSIM, and SIQE, and lower LPIPS indicate better quality. As shown in Fig. 5, our method achieves the best SIQE and LPIPS scores, and is second in PSNR and SSIM (w/o mesh is highest). This is partly because existing methods, including ours, compute PSNR and SSIM only over the common regions, due to the lack of ground truth for full stitched images. The w/o mesh variant achieves high alignment in these regions by allowing unrestricted deformation, which often causes severe distortion elsewhere. In contrast, our mesh constraint avoids excess deformation, ensuring better structure preservation and seamless results in the stitched output. Thus, a slightly lower PSNR and SSIM is reasonable, and our method achieves a better balance between quantitative scores and actual visual quality.
>
> ### Question 1: Depth Consistency and Improvement
> **Response:**
> - **[Depth consistency]** The depth estimation adopted in our work is based on direct estimation from a single image. As such, the resulting depth map only reflects the relative depth variations within the depicted scene, rather than representing the true absolute depth of the scene. Therefore, after obtaining depth maps from two different viewpoints, we perform a normalization operation (as described in lines 152 and 181 of the submitted manuscript) to unify the depth values across both maps. This ensures depth consistency for the same object across different viewpoints, which is essential for the accurate transformation estimation and ensuring the multi-view alignment precision.
>
> - **[Performance improvement]** In Fig. 6, the results obtained from w/o $L_{smooth}$, w/o $L_{cost}$, w/o $L_{reg}$ and w/o $L_{depth}$ exhibit structural breaks in the door area and label ghosting, while the result from w/o $L_{mesh}$ shows wall discontinuities. In contrast, our method achieves clear and accurate stitching results, demonstrating significant advantages over the ablated versions.
> Furthermore, in Table 2, our method achieves the best performance in SIQE and LPIPS metrics, while it is only slightly outperformed by w/o $L_{mesh}$ in PSNR and SSIM. Since ground-truth stitched images are unavailable, PSNR and SSIM are calculated only on the overlapping regions. The w/o mesh method allows excessive deformation for higher alignment in these areas, often distorting non-overlapping regions. In contrast, our mesh constraint limits such deformations and prevents unreasonable distortion. Thus, our slightly lower PSNR and SSIM are reasonable.
>
> Since the goal of image stitching is to ensure seamless results while preserving the integrity and correctness of the original structures, excessive pursuit of alignment in the common region at the cost of unwanted distortions is undesirable. In summary, considering both the visual and quantitative results, our method is more reasonable and effective compared to other ablated versions.
> ### Question 2: Architecture and Algorithm
> **Response:**
>
> - **[Soft-Seam estimation]** Due to space limitations, kindly refer to the first paragraph of our response to weakness 2 for further details.
>
> - **[RBA algorithm]** In the transformation regression process, we propose the Reparameterization Regression(RR) to enhance the network's fitting ability and prevent issues such as overfitting. As shown in Fig. 1(c), for RR, we use RepBlock as the basic operation block during the reparameterization process. Based on the proposed RBA algorithm, RepBlocks can be adaptively applied by pruning the 1×1 branch of the RepBlock according to its contribution. This approach helps to prevent noise or overfitting risks introduced during training. As a result, the model is adaptively simplified while avoiding redundant computations or harmful feature interactions.
>
> ### Question 3: Quantitative Improvement
> **Response:** In image stitching, it is essential to prevent issues such as ghosting, structural distortion, and misalignment after integrating scene content from different viewpoints. These problems serve as key factors in assessing the effectiveness of various stitching algorithms. However, the regions where such issues occur only occupy a small portion of the total pixel count in the generated wide-view images. Consequently, when evaluating performance using metrics that are calculated on a per-pixel basis, such as PSNR and SSIM, there is usually only a small difference among the results of different methods. This also explains why the values of different versions reported in Table 2 are relatively close. Despite this, problems like ghosting, structural distortion, and misalignment are highly perceptible in visual performance, leading to significant differences in visual quality between different methods, as shown in Fig. 6.
>
> ### Question 4: Reparameterization Design
> **Response:** RepBlock adds an extra auxiliary 1×1 convolution branch to the original 3×3 convolution branch, which provides improved feature diversity during training. During inference, these two branches are reparameterized into a single-path 3×3 convolution structure. This approach enhances feature representation capability, but the multi-branch structure can introduce complex coupling during the training process and may result in unstable optimization. In the image stitching task, convolutional layers for edge sensitivity and fine structure localization within our RR module may not require such structural complexity. Using RepBlock in these layers can introduce unnecessary noise or increase the risk of overfitting. As a result, not every convolutional layer equally benefits from this architectural complexity. To address this issue and improve performance while eliminating structural redundancy, we propose the RBA algorithm. This algorithm dynamically chooses between RepBlock and a standard 3×3 convolutional layer. By evaluating the contribution of the 1×1 and 3×3 branches in each RepBlock, the RBA algorithm prunes the 1×1 branch if it proves to have negative effects on feature extraction. Experimental results in Figure 9 demonstrate that selectively applying RepBlocks to intermediate and high-level semantic layers leads to better performance than applying them throughout the entire network.

---

> ### Author Response · Authors · 2025-08-05
>
> **We sincerely appreciate the time and thoughtful consideration you've dedicated to reviewing our work.**
>
> We're grateful for the opportunity to address your concerns through the detailed clarifications we provided, particularly regarding **network architecture** , **formulation explanation** , **analysis of both qualitative and quantitative results** , as well as a clarification of our paper’s **main contributions**, including depth prior improvement, soft-seam region based fusion, and the reparameterization strategy.
>
> We would be deeply grateful to know **whether our responses have satisfactorily addressed your questions and concerns**. Should you have any further points you'd like us to clarify during the remaining discussion period, we would be delighted to provide additional information or conduct any necessary analyses.
>
> **Thank you again for your constructive feedback and for helping us improve our research**.

---

> > ### Comment · Reviewer_Hpo2 · 2025-08-06
> >
> > The rebuttal has addressed my concerns well. If the authors include the clarifications and details in the camera-ready version, I am willing to raise my score.

---

> > > ### Author Response · Authors · 2025-08-06
> > >
> > > We sincerely appreciate your positive feedback and valuable suggestions. **We promise to include the clarifications and details mentioned in the rebuttal in the camera-ready version**. Your comments are very important to us and have greatly contributed to improving the quality of our paper. Thank you again for your thoughtful review.

---

### Official Review · Reviewer_H8ZG · 2025-07-02

**Clarity:** 3
**Significance:** 4
**Originality:** 3
**Rating:** 5
**Confidence:** 5

**Summary:**

This paper proposes a depth-supervised image stitching method, aiming to solve the stitching challenge of images with depth differences. This method achieves precise alignment of targets at different depths through two-stage deformation estimation combined with depth consistency constraints. Meanwhile, the strategy is determined through the soft seam area, and smooth fusion is achieved by combining the deep continuity constraint to alleviate the problems of ginning and misalignment. The re-parameterization strategy of shift regression further improves adaptability and reduces computational overhead.

**Questions:**

1. In cases where the input image exhibits no significant depth variation, is it still possible for this method to maintain accuracy?
2. In figure2, why is the color difference at the junction of the second group of stitched images larger compared to some comparison methods? What could be the possible reasons for this phenomenon? Has this type of image been evaluated in a user study, and what were the rating results?
3. Will the method introduced in this paper be made open-source? No relevant links or descriptions have been identified thus far.

**Ethical Concerns:**

["NO or VERY MINOR ethics concerns only"]

**Final Justification:**

My concerns are well addressed. I will keep my score.

**Limitations:**

Yes

**Paper Formatting Concerns:**

There are no major formatting issues.

**Quality:**

4

**Strengths And Weaknesses:**

Strengths:
1. The introduction of deep supervision effectively addresses the influence of depth disparities on image alignment, showcasing a robust solution to this critical challenge.
2. The soft seam fusion strategy significantly mitigates ghosting artifacts, resulting in superior visual quality and a more aesthetically pleasing output.
3. The reparameterization approach minimizes computational overhead while enhancing the overall performance of the method, demonstrating its efficiency.
4. The incorporation of depth maps to enhance the quality of image stitching represents a promising avenue for future research, highlighting its potential to advance the field.

Weaknesses:
1. It seems that the method proposed in this paper may have relatively high requirements regarding the generation quality of depth maps, which could potentially affect its applicability in certain scenarios. Could you provide more specific details on this aspect?
2. Given the complexity of the depth estimation method, there may be notable challenges in optimizing it to meet the demands of real-time applications.

---

> ### Author Rebuttal · Authors · 2025-07-30
>
> ## Summary of the reviews
>
> We sincerely thank the reviewer for their thoughtful evaluation of our work. We appreciate the recognition of our paper’s strengths in: (1) the incorporation of depth maps, (2) the design of re-parameterization, and (3) the improvement in stitching performance brought by soft seam region fusion. We acknowledge the reviewer's concerns about (1) the performance in scenarios with insignificant depth variations, and (2) the improvement of quantitative results. We will address these concerns in the following.
>
> ### Weakness 1: Dependence on Depth Quality
> **Response:** We would like to clarify that the proposed method improves multi-view alignment accuracy through the use of depth constraints and enables flexible fusion of multi-view scenes via soft- seam region localization. The combined effect of these two stages allows for accurate and realistic wide field-of-view reconstruction. Considering the limitations of monocular depth estimation, our method applies regularization to the depth maps from different viewpoints to ensure depth consistency for the same target across views. Therefore, any errors introduced due to the limited quality of the generated depth maps do not directly affect multi-view alignment performance. Furthermore, the subsequent fusion in the soft seam region further mitigates the impact of depth errors, ensuring the accuracy of the final stitching results.
>
> ### Weakness 2: Real-Time Capability
> **Response:** Thanks for your feedback. We acknowledge that, at present, our method’s computational efficiency does not fully meet the demands of real-time applications. Addressing real-time performance is an important direction for our future research, and we plan to further optimize our algorithm to improve its efficiency for practical use. Possible strategies include model compression, network pruning, and quantization, as well as leveraging hardware acceleration and more lightweight network architectures. We believe that with these targeted improvements, the proposed approach can be effectively adapted to scenarios requiring higher computational speed.
>
> ### Question 1: Performance with Insignificant Depth Variations
> **Response:** When the input images do not exhibit significant depth variation, it indicates that the objects in the scene lie at similar depths. Under such conditions, depth constraints remain valid, although their effect may be less pronounced than in scenes with strong depth variation. In addition to the depth constraint, the subsequent soft seam constraint continues to ensure the effectiveness of our method. For example, as shown in the second group of images in Figure 1 of the supplementary materials, the stitched image contains multiple planes with similar depths. In this case, the fine sign structures are barely distinguishable in the estimated depth maps. Nevertheless, our method still achieves successful stitching.
>
> ### Question 2:  Color Difference
> **Response:** In this scenario, there are noticeable brightness differences between the target and reference images due to variations in shooting angles. Our method provides gradual compensation for color differences within the fusion regions during multi-view scene integration. However, we do not perform color adjustment across the entire image to avoid negatively affecting the color accuracy of other objects in the scene. As a result, compared to other approaches, our soft-seam based flexible fusion achieves smoother transitions in the multi-view overlap areas and effectively avoids abrupt color changes.
>
> During user studies, participants were asked to evaluate quality from multiple perspectives, including ghosting, misalignment, structural accuracy, and realistic scene restoration. Therefore, the results of the user study represent a comprehensive assessment of all these factors. Compared with other methods, the proposed method demonstrates superior overall performance based on this comprehensive evaluation.
>
>
> ### Question 3: Open Source
> **Response:** Due to the rebuttal policy, we are not allowed to provide open-source links. We will release the code and resources as soon as the paper is accepted, to facilitate comparison and use by the researchers.

---

> > ### Comment · Area_Chair_AdWe · 2025-08-06
> >
> > Dear Reviewer,
> > Could you please read the rebuttal, update your final review, and share any remaining follow-up questions, if any? Also, kindly acknowledge once this is done.
> > Thank you.

---

> > ### Comment · Reviewer_H8ZG · 2025-08-07
> > **Thank you for your response.**
> >
> > Thank you for your response. Regarding weakness 1, based on the author's explanation it seems the proposed method does not strongly rely on depth quality. In that case, can this module be replaced with a faster and simpler estimation approach to further improve computational efficiency?

---

> > > ### Author Response · Authors · 2025-08-07
> > >
> > > Our method achieves precise image stitching through depth-prior guidance and soft-seam region fusion. A higher-quality depth map is therefore expected to enhance the entire pipeline. We evaluated three depth-estimation networks (Monodepth [1], Marigold [2] and DepthAnything) and supplied their predicted depth maps to the stitching procedure. The quantitative results are listed below:
> > >
> > > | Method                        | PSNR(↑)|SSIM(↑)| SIQE(↑)|LPIPS(↓)|
> > > |------------------|--------|-------|--------|-------|
> > > | **Monodepth2-Based**| 24.088 | 0.893 | 43.767 | 0.422 |
> > > | **Marigold-Based** | 24.081 | 0.891 | 43.781 | 0.427 |
> > > | **Ours(DepthAnything-Based)** | 24.187 | 0.897 | 43.783 | 0.416 |
> > >
> > > We can see that DepthAnything yields the highest stitching accuracy. To assess both accuracy and computational efficiency, we compared the inference times of different strategies on a system equipped with an Intel Core i7-13700KF 3.40 GHz CPU and an NVIDIA RTX 3090 GPU. As illustrated below, DepthAnything offers a significantly lower computational cost compared to the others. Taken together, these findings justify the use of DepthAnything as the depth estimation module in our system.
> > >
> > > | Method                        | Time(ms)|
> > > |-------------------------------|---------|
> > > | **Monodepth2-Based**          |  15.757 |
> > > | **Marigold-Based**            | 197.341 |
> > > | **Ours(DepthAnything-Based)** |   8.483 |
> > >
> > > [1] Digging into Self-Supervised Monocular Depth Prediction. ICCV 2019.
> > > [2] Repurposing Diffusion-Based Image Generators for Monocular Depth Estimation. CVPR 2024.
> > >
> > > We hope that the above experiments comparing different depth estimation algorithms in terms of stitching performance and efficiency address the concerns you may have.

---

> > > > ### Comment · Reviewer_H8ZG · 2025-08-09
> > > > **My concerns are well addressed.**
> > > >
> > > > Thank you for the reply. My concerns are well addressed. I will keep my score.

---

### Official Review · Reviewer_xeFo · 2025-07-06

**Clarity:** 3
**Significance:** 3
**Originality:** 2
**Rating:** 4
**Confidence:** 5

**Summary:**

This paper presents a depth-supervised, seam-free image stitching framework tailored for large-parallax scenarios. It introduces a depth-aware two-stage transformation estimation, a soft-seam multi-view fusion module for seamless transitions, and a reparameterization-based shift regression strategy to improve computational efficiency. Experiments show that this method has certain effectiveness.

**Questions:**

Please refer to the weaknesses proposed in the strength and weakness section.
And I have some minor questions:
1. What are the time and space complexities of the proposed method?
2. All experiments in the paper are conducted on stitching two images. However, real-world applications such as 360° panoramic stitching often involve stitching 5 to 10 images and require addressing challenges like loop consistency and global error propagation. I’m curious about how the proposed method performs in such scenarios.

**Ethical Concerns:**

["NO or VERY MINOR ethics concerns only"]

**Final Justification:**

My concerns are well addressed and I will keep the score.

**Limitations:**

Yes

**Quality:**

3

**Strengths And Weaknesses:**

Strengths
1. The proposed depth-aware two-stage transformation estimation module effectively mitigates alignment issues caused by depth variations. The motivation is clear, and the design is well-justified.
2. The soft-seam multi-view fusion module relaxes the definition of stitching seams, significantly improving the naturalness of image transitions.
3. The ablation studies are thorough and systematic, clearly demonstrating the contributions of each module and loss component.

Weaknesses
1. The $L_{depth}$ relies entirely on the depth maps provided by Depth Anything. However, as a general-purpose model not specifically designed for image stitching tasks, Depth Anything may introduce biases from its training data. Consequently, the loss may misfire in regions with noisy or discontinuous depth maps. It can penalize correct predictions, introduce misleading supervision signals, and steer the main model in the wrong direction.
2. All experiments are conducted on standard-sized images, whereas real-world stitching tasks often involve ultra-high-resolution images or multi-scale inputs. The paper lacks experiments under such practical and challenging scenarios.
3. The paper does not include comparisons with image stitching methods based on generative models[1], which are becoming increasingly relevant in recent research.
4. The paper uses a large number of symbols in its equations, but not all of them are clearly defined. For example, $p_k$ in Equ. (2). It is recommended that the authors include a table of symbol definitions to improve readability.
5. There are some grammatical errors in the paper, such as "a depth-supervised image stitching" in line 43 and "state-of-the-arts" in line 64. It is recommended that the authors carefully review and correct the grammatical issues throughout the paper.

[1] Recdiffusion: Rectangling for image stitching with diffusion models. CVPR 2024.

---

> ### Author Rebuttal · Authors · 2025-07-30
>
> ## Summary of the reviews
>
> We sincerely thank the reviewer for their thoughtful evaluation of our work. We appreciate the recognition of our paper's strengths in: (1) clear motivation and well-justified design, (2) improved transition naturalness enabled by the soft-seam fusion module, and (3) thorough analysis and superior performance of our method. We acknowledge the reviewer's concerns about (1) the impact of depth estimation quality on the stitching performance, (2) the effectiveness of our method when handling images of different resolutions; and (3) the performance in panoramic stitching. We will address these concerns in the following.
>
>
> ### Weakness 1: Depth Anything for Depth Supervision
> **Response:** The goal of image stitching is to realistic and accurate wide-field reconstruction results. Therefore, multi-view alignment and multi-view fusion are the two core steps in the stitching task. In our method, to improve the accuracy of multi-view alignment, we introduce depth constraints provided by Depth Anything. However, since the Depth Anything model performs inference based on the single image, it may result in inconsistent depth values for the same object under different viewpoints. To this end, we regularize the depths from different views to ensure the depth consistency of the same object across multiple perspectives. In addition, to avoid the impact of depth bias on alignment errors and, consequently, on overall stitching performance, we propose a soft-seam region localization during the multi-view fusion phase. Instead of restricting the stitching seam to a fixed line, we extend it to a candidate region and calculate the optimal fusion within this area by cost evaluation. This further improves the realism and accuracy of the stitching results.
>
> Therefore, our method ensures realistic and accurate stitching outcomes through the combined effect of depth-based constraints and optimal soft-seam fusion. Even when the depth maps fail, the subsequent soft-seam fusion struggles to provide a certain degree of correction. For example, as shown in the third example in Figure 1 of the supplementary material, although the generated depth maps are not ideal in this case, our method still achieves satisfactory stitching results, demonstrating the tolerance for the depth error.
>
> ### Weakness 2: Performance Across Different Resolutions
> **Response:** Thanks for your suggestion. We conducted performance evaluations on both low-resolution images (32×32) and high-resolution images (1024×1024), with both types processed from the IVSD dataset. Quantitative comparisons are shown in the table below, where the best results are highlighted in bold. We can see that, the proposed method is able to achieve outstanding performance even on data with extremely high or extremely low resolutions.
> | Size      | Metric | APAP  | ELA   | LPC   | SPW   | UDIS  | UDIS++ | TRIS  | SRS   | **Ours** |
> |-----------|--------|-------|-------|-------|-------|-------|--------|-------|-------|----------|
> | **32×32** | PSNR(↑)  | 22.551| 21.184| 22.915| 22.482| 24.618| 23.481 | 23.178| 22.481| **25.735**|
> |           | SSIM(↑)  | 0.726 | 0.717 | 0.764 | 0.805 | 0.755 | 0.722  | 0.804 | 0.815 | **0.882** |
> |           | SIQE(↑)  | 39.385| 38.843| 38.587| 37.433| 40.185| 42.013 | 43.465| 43.018| **44.627**|
> |           | LPIPS(↓) | 0.453 | 0.422 | 0.473 | 0.553 | 0.502 | 0.419  | 0.418 | 0.442 | **0.402** |
> | **1024×1024**| PSNR(↑)| 24.218| 25.354| 24.368| 24.352| 25.184| 24.389 | 23.171| 22.481| **26.101**|
> |           | SSIM(↑)  | 0.775 | 0.792 | 0.815 | 0.781 | 0.818 | 0.833  | 0.801 | 0.831 | **0.878** |
> |           | SIQE(↑)  | 42.433| 42.732| 41.834| 40.384| 43.152| 43.839 | 42.501| 43.437| **44.143**|
> |           | LPIPS(↓) | 0.444 | 0.432 | 0.465 | 0.487 | 0.415 | 0.425  | 0.422 | 0.416 | **0.410** |
>
> ### Weakness 3: Evaluation on Recdiffusion
> **Response:** We have conducted a detailed investigation on Recdiffusion. This paper is specifically designed for rectangling stitched images. Therefore, we can compare the performance of different stitching algorithms by evaluating their results within the Recdiffusion framework. Specifically, we first selected from the DIR-D dataset (used by Recdiffusion) those image groups that correspond to those in the UDIS-D dataset, obtaining both input images and ground truth. We then manually constructed the required input and ground truth data pairs. In our experiment, we first obtained stitching results using different algorithms, and then assessed their rectangling performance using Recdiffusion. In addition to the commonly used PSNR and SSIM metrics, we also incorporated the FID metric to evaluate image quality. The results are shown in the table below. It can be observed that our method provides better support for the rectangling task compared to other methods.
> | Metric | APAP  | ELA   | LPC   | SPW   | UDIS  | UDIS++ | TRIS  | SRS   | **Ours** |
> |--------|-------|-------|-------|-------|-------|--------|-------|-------|----------|
> | PSNR(↑)  | 18.24 | 19.48 | 17.18 | 16.73 | 18.67 | 17.76 | 18.49 | 19.73 | **21.18** |
> | SSIM(↑)  | 0.661 | 0.673 | 0.701 | 0.586 | 0.671 | 0.627 | 0.677 | 0.691 | **0.733** |
> | FID(↓)  | 8.476 | 5.011 | 6.373 | 7.368 | 5.136 | 6.199 | 8.411 | 7.157| **4.347** |
>
> ### Weakness 4: Symbol Readability
> **Response:** Thank you for your valuable suggestion. We have carefully reviewed all formulas and provided clear definitions for each symbol, as shown below. Additionally, we will include this symbol table in the paper to further improve readability.
> | Symbol | Definition |
> | --------- | --------- |
> | $I_{r}$ | Reference image |
> | $I_{t}$ | Target image |
> | $I_{wr}$ | Warped reference image |
> | $I_{wt}$ | Warped target image |
> | $M_{wr}$ | Warped reference image mask |
> | $M_{wt}$ | Warped target image mask |
> | $M_{sr}$ | Stitched reference image mask |
> | $M_{st}$ | Stitched target image mask |
> | $M_{s}$ | Seam area mask |
> | $I_{s}$ | Stitched image |
> | $p_k$ | The coordinate of the k-th point in the reference image |
> | $p_k'$ | The coordinates of $p_k$ at the corresponding point of the target image |
> | $I_{wdr}$ | Warped reference depth image |
> | $I_{wdt}$ | Warped target depth image |
> | $\odot$ | Pixel-by-pixel multiplication |
> | $\neg M$ | Reverse and expand the mask |
>
> ### Weakness 5: Grammatical Errors
> **Response:**  We appreciate your careful review. We have corrected the errors such as “a depth-supervised image stitching” and “state-of-the-arts” as suggested. Additionally, we have thoroughly reviewed the entire manuscript to address and correct other grammatical problems to improve the overall clarity and readability.
>
> ### Question 1: Time and Space Complexity
> **Response:** We have tested the time and space complexities of the proposed method. The computational cost is 125.8 GFLOPs and the memory access is 2624 MB.
>
> ### Question 2: Performance on Panorama Stitching
> **Response:** To evaluate the performance of our method in 360° panoramic image stitching tasks, we conducted experiments on the MSGA [1] dataset. This dataset was captured using a binocular imaging device capable of infrared and visible imaging across various scenarios, encompassing a wide range of 360° scene content. In our experiments, we selected the visible images for panoramic stitching validation, with each scene containing 5 to 15 consecutive images. Since most existing image stitching algorithms support only two images as input, we adopted a pairwise stitching strategy. Specifically, we first stitch the first and second images, then use the stitched result as the reference image to stitch with the third image, and so on, until all images in the scene are stitched together. To ensure fairness, none of the methods involved additional processing such as loop closure verification or global cumulative error correction during the experiments.
>
> The quantitative comparison results of various methods are shown in the table below. Your suggestions have provided important insights for our work. In the future, we will further optimize our method design by incorporating your suggestions, with particular focus on addressing key challenges such as loop consistency and global error propagation in panoramic stitching, so as to continuously improve the robustness and applicability of our method to multi-input panoramic stitching. Once again, thank you for your valuable feedback.
>
> [1] Jiang Z, Zhang Z, Liu J, et al. Multispectral image stitching via global-aware quadrature pyramid regression[J]. IEEE Transactions on Image Processing, 2024.
>
> | Metric | APAP  | ELA   | LPC   | SPW   | UDIS  | UDIS++ | TRIS  | SRS   | **Ours** |
> |--------|-------|-------|-------|-------|-------|--------|-------|-------|----------|
> | PSNR(↑)  | 20.167| 22.439| 21.697| 22.738| 22.133| 22.735 | 21.953| 22.921| **23.104**|
> | SSIM(↑)  | 0.735 | 0.791 | 0.733 | 0.805 | 0.767 | 0.791  | 0.752 | 0.801 | **0.811** |
> | SIQE(↑)  | 38.499| 39.714| 37.498| 39.134| 38.125| 38.326 | 37.053| 38.135| **40.137**|
> | LPIPS(↓) | 0.473 | 0.427 | 0.481 | 0.453 | 0.478 | 0.455  | 0.439 | 0.475 | **0.410** |

---

> > ### Comment · Reviewer_xeFo · 2025-08-05
> >
> > Thank you for the reply. My concerns are well addressed.

---

### Decision · Program_Chairs · 2025-09-17

**Decision:**

Accept (poster)

**Comment:**

The paper received positive reviews (2 borderline accept, 1 accept, 1 strong accept).

The reviewers praised the approach (the two stage deformation estimation and stitching with depth consistency constraints) finding that the results were strong. Reviewers suggested that the breadth of the experiments could be expanded (such as 360 pano stitching) as well as more thorough comparisons (such as generative techniques).

I also advocate for acceptance but would strongly encourage the authors to edit the manuscript in adherence with the suggestions of Reviewer xeFo and Reviewer Hpo2.